# Development of a Japanese Version of the Formula for Calculating Periodontal Inflamed Surface Area: A Simulation Study

**DOI:** 10.3390/ijerph19169937

**Published:** 2022-08-11

**Authors:** Haruka Ueda, Norio Aoyama, Shinya Fuchida, Yuki Mochida, Masato Minabe, Tatsuo Yamamoto

**Affiliations:** 1Department of Dental Sociology, Kanagawa Dental University, Yokosuka 238-8580, Japan; 2Department of Periodontology, Kanagawa Dental University, Yokosuka 238-8580, Japan; 3Department of Education Planning, Kanagawa Dental University, Yokosuka 238-8580, Japan; 4Bunkyo Do-ri Dental Clinic, Chiba 263-0024, Japan

**Keywords:** periodontal inflamed surface area (PISA), Japanese, inflammatory burden, periodontitis, body mass index, simulation, periodontal epithelial surface area

## Abstract

The periodontal inflamed surface area (PISA) is a useful indicator of periodontal status. However, its formula was based on a meta-analysis involving five countries, and racial differences in tooth root morphology could have affected the calculations. This study aimed to develop a Japanese version of the PISA and compare it with the original version. The formulas reported by a previous Japanese study calculating the amount of remaining periodontal ligament from clinical attachment measurements were used to calculate the PISA. A simulation was performed to compare the Japanese version with the original version by inputting probing pocket depth (PPD) from 1 to10 mm and by using clinical data. The PISA values in the Japanese version were larger and smaller than those in the original version for PPDs of 1–5 mm and 6–10 mm, respectively. The PISA values for the clinical data from the Japanese version were significantly higher than those from the original version. Both versions of the PISA values correlated equally well with body mass index. The Japanese version of the PISA can be used to assess the amount of inflamed periodontal tissue resulting from periodontitis in Japanese populations, taking into account racial heterogeneity in root morphologies.

## 1. Introduction

Periodontitis is a chronic inflammatory disease that breaks down the tooth-supporting structures. Indicators such as the clinical attachment level (CAL), probing pocket depth (PPD), bleeding on probing (BOP), and tooth mobility are routinely used for clinical assessment and/or surveys of periodontal diseases. It has been proposed that the periodontal inflamed surface area (PISA) determines the total periodontal inflammatory status in periodontitis patients [1]. The PISA is calculated by using BOP, combined with either PPD or CAL, and gingival recession to quantify the bleeding pocket epithelial surface area, measured in square millimeters, for all teeth.

The PISA has some advantages in data processing and analysis because it can be treated as a continuous variable to quantify periodontal inflammation [2]. The PISA is strongly correlated with the percentage of BOP, and it reflects treatment efficacy more sensitively than BOP [3]. Moreover, the PISA is positively correlated with glycosylated hemoglobin A1c (HbA1c) levels in type 2 diabetes mellitus patients [4,5] and high-sensitivity C-reactive protein (hsCRP) levels in end-stage renal disease patients [6] and healthy adults [7], as well as body mass index (BMI) in dental patients [8]. These reports suggest that the PISA can be a useful indicator for dentists to share information about periodontal disease with medical doctors and/or patients.

Although the PISA has certain advantages, its calculation is based on a formula derived from the results of a meta-analysis involving the normal ranges for the root surface areas of the permanent dentition [9], published mean values for root length [10], and the results of a study associating the percentage of root surface area to the percentage of the remaining root length [1,11]. Intriguingly, the studies were based on a meta-analysis involving countries with diverse racial backgrounds, such as Japan, India, England, Russia, and the United States [9]. These studies showed that the root-morphology-associated factors, such as the number of roots in permanent molars [12,13] and crown-to-root ratios [14], differed by ethnicity; hence, it is worthwhile considering a modified version of the original formula to determine the PISA taking into account the existing racial differences in root morphology. Therefore, the present study aimed to develop a Japanese version of the formula for calculating the PISA using measurements of the remaining periodontal ligament derived from CAL measurements [15] and to compare it with the original version of the formula.

## 2. Materials and Methods

The development of a Japanese version of the formula for calculating periodontal inflamed surface area was conducted in three phases. In Phase I, formulas quantifying the periodontal epithelial surface area (PESA) of 14 tooth types except the third molars were generated. In Phase II, the Japanese version of the formula for deriving the PISA and the original version were compared by simulation. In Phase III, the validity of the Japanese version of the PISA was tested by comparison of the PISA from the two versions using clinical cross-sectional data.

### 2.1. Phase I: Japanese Version of the Formula Calculating the PESA and the PISA

Formulas quantifying the PESA of 14 tooth types except the third molars were generated using the results of a previous study in Japan [15]. Extracted teeth used in the previous study were selected randomly from more than 20,000 extracted teeth collected from 130 dentists across Japan.

The term “periodontal epithelial surface” encompasses any sulcular or pocket epithelium. The Japanese version of the formula for calculating the PESA was formulated using the slope of a linear function that estimated the amount of remaining periodontal supporting tissue by measuring the attachment levels in the extracted teeth of Japanese patients (Table 1) [15], i.e., the value of the slope determined the PESA for each 1 mm length of attachment level or PPD. Conversely, the original version of the formula for determining the PESA uses the coefficients of six polynomials (Table 1) [1,16].

Since the PESA includes the area of the pocket epithelium both with and without inflammation, the number of BOP-positive sites should be considered when calculating the PISA. Therefore, the PISA of each tooth was calculated by multiplying the measured PESA by the percentage of BOP-positive sites on each tooth. Subsequently, the sum of the PISA measurements of all teeth was calculated.

### 2.2. Phase II: Comparison of the PISA Values Obtained from the Two Versions by Simulation

The Japanese version of the formula for deriving the PISA and the original version were compared in the case of PPD of 1 to 10 mm with all sites exhibiting BOP positivity, in the case of PPD of 1 mm with BOP positivity on all six sites around the tooth for all teeth except the third molars, PPD of 2 mm with BOP positivity on all six sites around the tooth for all teeth except the third molars, and so on. Furthermore, a comparison was made of the calculated values obtained using both versions for each tooth type and the sum of all teeth except the third molars.

### 2.3. Phase III: Comparison of the PISA Values Obtained from the Two Versions Using Clinical Data

#### 2.3.1. Participants

The data from our previous clinical study [8] showing the association between periodontal status and obesity were used to compare the two versions of the PESA and the PISA. Of the data from the 235 participants, those from 25 subjects were excluded because of missing data for PPD and/or BOP. Data for PPD and BOP of 210 subjects (mean age: 76.7 years, standard deviation (SD) of age: 12.4 years; males 70, females 140) were used for the analysis.

#### 2.3.2. Statistical Analysis

The PESA and the PISA were calculated and compared between the two versions using the Wilcoxon signed-rank test. Because the previous study showed a significant positive association between the PISA and BMI, the correlation of the PISA of each version and BMI was assessed using Spearman’s rank correlation coefficient. Then, multiple regression analysis was used to assess the correlation of the PISA of each version and BMI after adjusting for age and sex. A sample size of more than 200 was considered sufficient, since multiple regression analysis with six explanatory variables was performed with 72 subjects in a similar previous study [17]. The statistical analyses were performed using IBM SPSS Statistics 27.0 for Windows (SPSS Japan Inc., Tokyo, Japan) with a significance level of 5%.

#### 2.3.3. Ethical Approval

This study was approved by the Ethics Committee of Kanagawa Dental University (No. 835, 29 March 2022) and conducted in full accordance with the guidelines set forth by the World Medical Association Declaration of Helsinki. Written informed consent was provided by all participants prior to participation in the previous study [8] (No. 665, 13 May 2020; No. 801, 19 October 2021). All participants were informed that their data would be used for the present study (opt-out system).

## 3. Results

### 3.1. Phase I: Calculation of the PISA

A Microsoft Excel spreadsheet was constructed to formulate a Japanese version of the PISA after calculating the PESA. The following steps were used in the calculation:After filling in the PPD measurements for six sites per tooth in the spreadsheet, the computer calculated the mean PPD value for each tooth type.The mean PPD value around a particular tooth was entered into the appropriate formula (Table 1) [15] for the translation of linear PPD measurements to the PESA for that specific tooth. For example, if the probing pocket depth is 5 mm on a maxillary central incisor, the PESA is 16.96 × 5 or 84.8 mm^2^ (Table 1).The sum of all individual PESA values around individual teeth was calculated.BOP positivity or negativity for each measurement site was entered as a value of 1 or 0, respectively, in the appropriate column of the spreadsheet.The PESA value for a particular tooth was then multiplied by the proportion of sites around the tooth with BOP. For example, if two of the maximum six sites were BOP positive, the PESA of that particular tooth was multiplied by 2/6, thereby deriving the PISA for that specific tooth.The sum of all individual PISA values around the individual teeth was finally calculated, amounting to the total PISA within a patient’s oral cavity.

The spreadsheets are freely available as a Appendix A. All it takes to calculate the PISA is filling in the PPD and BOP values for six sites per tooth in the freely downloadable spreadsheet.

### 3.2. Phase II: Comparison of the PISA from the Two Versions by Simulation

Figure 1 shows the simulated PISA of each tooth type except the third molar when the PPD varied from 1 to 10 mm and with BOP positivity on all six sites around the tooth in the maxilla. The PISA values of the central and lateral incisors, canine, and first premolar were similar between the two versions. When the PPD value ranged from 1 to 6 mm, the Japanese version of the PISA for the second premolar showed smaller values than the original version. When the PPD value ranged from 1 to 5 mm, the PISA values for the first molar derived from the Japanese version showed larger values than the original version. When the PPD value ranged from 6 to 10 mm, the PISA values for the first and second molars derived from the Japanese version showed smaller values than the original version.

Figure 2 shows the simulated PISA of each tooth type except the third molar in the mandible. The PISAs of the lateral incisor and the first premolar were similar between the two versions from PPDs of 1 to 8 mm. The PISAs of the central incisor, canine, second premolar, and first molar were smaller in the Japanese version than in the original version. When the PPD value ranged from 1 to 4 mm, the Japanese version of the PISA for the second molars showed larger values than the original version. However, when the PPD value ranged from 5 to 10 mm, the PISA values obtained from the Japanese version for the second molar showed smaller values than those obtained from the original version.

Figure 3 shows the simulated PISAs of all teeth except the third molars when the PPDs varied from 1 to 10 mm and all six sites around each tooth were BOP positive. When the PPD value was 1 to 5 mm, the PISA value of the Japanese version showed larger values than those obtained from the original version (for PPD = 3 mm, the Japanese version: 1879.6 mm^2^, the original version: 1616.1 mm^2^). When the PPD value was large, the PISA value of the Japanese version for the second molar showed smaller values than those obtained from the original version (for PPD = 8 mm, the Japanese version: 5012.2 mm^2^, the original version: 5382.8 mm^2^). The two lines crossed between PPD = 5 and 6 mm.

### 3.3. Phase III: Comparison of the PISA Values from the Two Versions Using Clinical Data

Table 2 shows the characteristics of the subjects.

The comparisons of the PESA or PISA values from the two versions using clinical data are shown in Table 3. Both the PESA and PISA values obtained from the Japanese version were significantly larger than those obtained from the original version (*p* < 0.001).

The Spearman’s rank correlation coefficients of the PISA of the Japanese version and the original version with BMI were 0.190 (*p* = 0.006) and 0.192 (*p* = 0.005), respectively. The standardized coefficients of the PISA of the Japanese version and the original version after adjusting for age and sex were 0.204 (*p* = 0.003) and 0.192 (*p* = 0.006), respectively.

## 4. Discussion

Considering the existing racial differences in root morphology, a Microsoft Excel spreadsheet was constructed to formulate a Japanese version of the PISA using the slopes of the formulas that calculate the amount of remaining periodontal ligament from CAL measurements using extracted teeth in Japan [15]. It is noteworthy that the Japanese version is a linear function, whereas the original version is six polynomials, and the Japanese version has a simpler formula. In addition to research purposes, the spreadsheet may also be used to provide an explanation to patients and/or medical doctors regarding the surface area of bleeding pocket epithelium. To calculate the PESA and PISA for the third molars, the formula from the original version was used in the spreadsheet.

The results of the simulation showed differences in the Japanese version and the original version in both individual tooth PISA and total PISA values for all teeth. In particular, a difference was noted in molars that have larger root surface area than any other tooth types, and the heterogeneity resulted in the differences associated with the total PISA values obtained from both versions. For example, the PISA values for the Japanese version were larger than the values obtained from the original version for the maxillary first molars and mandibular first and second molars with PPD measurements from 1 to 5 mm. The differences reflect the larger values in the total PISA in the Japanese version over the original version with PPD from 1 to 5 mm. The difference in the total PISA for all teeth between the two versions was confirmed using clinical data: the PISA values obtained from the Japanese version were significantly larger than those from the original version. In addition, BMI was significantly correlated with the PISA values from both versions after adjusting for age and sex to the same degree, suggesting that the PISA values from the Japanese version reflect the amount of periodontal inflammation at least as much as those from the original version.

The difference in the total PISA, especially for PPD up to 5 mm and PISA up to 3000 mm^2^, is critical for evaluating the PISA value as precisely as possible because such values are commonly observed in clinical practice, as well as in epidemiological surveys [18]. A clinical study from South Korea reported mean (SD) PISA values of 116 (16), 260 (30), and 407 (94) mm^2^ for mild, moderate, and severe periodontal disease, respectively [2]. A clinical study from Spain showed that mean (SD) PISA values for the periodontally healthy group and mild, moderate, and severe periodontitis groups were 34 (16), 293 (98), 646 (86), and 2309 (588) mm^2^, respectively [19]. A retrospective clinical study assessing periodontal treatment in Japan reported median PISA values of 1271 mm^2^ at the first examination, 212 mm^2^ at the end of the initial preparation, 51 mm^2^ at the supportive periodontal therapy (SPT) transition, and 30 mm^2^ at the latest SPT [3]. Because the total PISA in the Japanese version for PPD from 1 to 5 mm was approximately 1.01 to 1.21 times higher than the original version’s PISA in the present study, accuracy can be improved by using the Japanese version for Japanese patients.

Although the simulated total PISA values in the Japanese version were larger than those in the original version, with PPD ranging from 1 to 5 mm, those in the Japanese version were smaller than those in the original version, with PPD ranging from 6 to 10 mm. The difference was mainly ascribed to the difference in the tooth-level PISA of the maxillary first and second molars and the mandibular second molar. The molars have larger root surface areas than any other tooth types, and a racial difference was noted in the molars; for example, in the mandibular second molar, the prevalence of a third root was 0.8% and 2.6% for the Asian and White groups, respectively [13]. Because the number of roots correlates positively with the root surface area, the morphological difference agrees with the results of the present study.

Although the Japanese version of the PISA has some advantages over the original version, the Japanese version and the PISA itself have some limitations, regardless of the version. First, the formula for each tooth type in the Japanese version was based on 420 extracted teeth (30 extracted teeth per each tooth type) [15]. Whether the selected teeth are representative of those of Japanese people as a whole is unclear. However, the teeth were selected randomly from more than 20,000 extracted teeth collected from 130 dentists across Japan [20], and the root length of each tooth type was similar to that reported previously, suggesting that the extracted teeth used in the previous study provide an accurate representation of each tooth type [15]. Second, as pointed out in the original version of the PISA [1], the PPD and BOP measurements used for the calculation of the PISA entail measurement errors attributable to the observer and the instruments. Third, individual variations in the root surface area and root length were not considered for the calculation of the PISA.

In addition to the limitations described above, the following are other concerns to be addressed: The multiple regression models did not include possible confounders including smoking status other than age and sex. Due to this shortcoming, the model might not accurately analyze the association between the PISA and obesity. Since the original purpose of the present study was to compare the original and Japanese versions of the PISA, if this shortcoming occurs to the same degree in both models, its impact can be seen as being of equal magnitude and may have little effect on the comparison. Moreover, the periodontal status of the participants was relatively healthy, and most had mild periodontal disease. Because of the usefulness of the PISA in moderate and severe periodontal disease, further studies with subjects having these periodontal conditions are warranted to clarify the usefulness of the Japanese version of the PISA compared to the original version.

## 5. Conclusions

The Japanese version of the PISA was developed to assess the inflamed periodontal epithelial surface area based on the formula calculating the amount of remaining periodontal ligament from the CAL measurement using extracted teeth in Japan, and it was compared to the original version using simulated PPDs from 1 to 10 mm with BOP positivity on all sites and clinical data. The PISA values of all teeth except the third molars in the Japanese version were larger and smaller than those in the original version for PPDs of 1 to 5 mm and 6 to 10 mm, respectively. The PISA values for the clinical data from the Japanese version were significantly higher than those from the original version. In addition, the PISA values calculated from both Japanese and original versions were significantly correlated with BMI to the same degree after adjusting for age and sex. These results showed that the Japanese version of the PISA can be used, taking into account the racial differences in the morphology of tooth roots.

## Figures and Tables

**Figure 1 ijerph-19-09937-f001:**
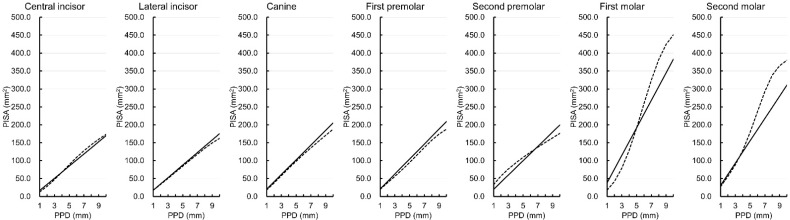
Simulated periodontal epithelial surface area (PISA) for probing pocket depths (PPDs) of 1 to 10 mm with bleeding on probing positivity at all six sites around the tooth in the Japanese version (solid lines) and original version (dotted lines) [1] for each tooth type in the maxilla.

**Figure 2 ijerph-19-09937-f002:**
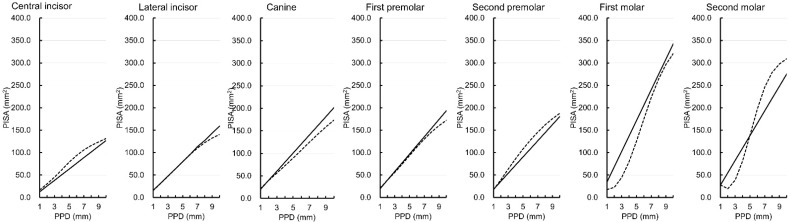
Simulated periodontal epithelial surface area (PISA) for probing pocket depths (PPDs) of 1 to 10 mm with bleeding on probing positivity at all six sites around the tooth in the Japanese version (solid lines) and original version (dotted lines) [1] for each tooth type in the mandible.

**Figure 3 ijerph-19-09937-f003:**
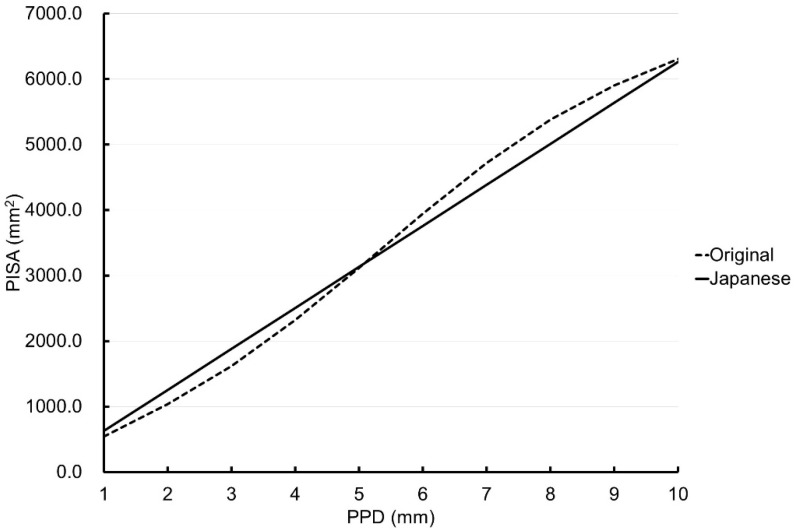
Simulated periodontal epithelial surface area (PISA) for probing pocket depths (PPDs) of 1 to 10 mm with bleeding on probing positivity at all six sites around the tooth in the Japanese version (solid lines) and original version (dotted lines) [1] in the whole mouth except the third molars.

**Table 1 ijerph-19-09937-t001:** Coefficients of the function of periodontal epithelial surface area (PESA) of the original version and the Japanese version for each tooth type.

Tooth Type	Japanese Version ^1^	Original Version ^2^					
		a_1_	a_2_	a_3_	a_4_	a_5_	a_6_
Maxilla							
Central incisor	16.96	12.3905	0.1374	0.6717	−0.14536	0.01126	−0.0003083
Lateral incisor	17.54	18.7571	−1.6471	0.5258	−0.07900	0.05890	−0.0001855
Canine	20.52	16.5369	1.6010	−0.2494	0.01087	0.00021	−0.0000182
First premolar	20.90	21.8618	−2.3031	0.5330	−0.04075	0.00062	0.0000199
Second premolar	19.98	39.2681	−7.3113	1.2340	−0.12192	0.00626	−0.0001260
First molar	38.33	16.8835	−0.5688	1.5433	−0.06519	−0.01454	0.0009019
Second molar	31.10	25.4265	4.6241	−3.0787	0.95774	−0.10923	0.0040876
Mandible							
Central incisor	12.70	21.4600	−6.6888	2.4638	−0.39094	0.02743	−0.0007116
Lateral incisor	15.91	16.4395	−1.0337	0.4146	−0.05711	0.00257	−0.0000211
Canine	20.13	24.6992	−3.5868	0.6903	−0.05799	0.00189	−0.0000142
First premolar	19.32	24.6866	−4.8531	1.3992	−0.18028	0.01037	−0.0002229
Second premolar	18.04	13.1705	5.0958	−1.0989	0.10864	−0.00559	0.0001179
First molar	34.27	19.1229	−12.2566	5.5750	−0.78145	0.04566	−0.0009711
Second molar	27.56	46.6148	−43.1558	16.7577	−2.48858	0.16174	−0.0038873

^1^ The value was the slope of a linear function to calculate the Japanese version of the PESA. For example, if the probing pocket depth is 5 mm on a maxillary central incisor, the PESA is 16.96 × 5 or 84.8 mm^2^. ^2^ The values of a_1_ through a_6_ were the coefficients of the polynomial a_1_x + a_2_x^2^ + a_3_x^3^ + a_4_x^4^ + a_5_x^5^ + a_6_x^6^ to calculate the original version of the PESA. For example, if the probing pocket depth is 5 mm on a maxillary central incisor, the PESA is 12.3905 × 5 + 0.1374 × 5^2^ + 0.6717 × 5^3^𢈒 0.14536 × 5^4^ + 0.01126 × 5^5^− 0.0003083 × 5^6^ or 88.9 mm^2^ [1].

**Table 2 ijerph-19-09937-t002:** Subjects’ characteristics (*n* = 210).

Variable	Median	25th Percentile	75th Percentile
Age (y)	70	61	76
Body mass index (kg/m^2^)	22.9	20.4	25.0
Number of teeth present	25	21	27
Mean probing pocket depth (mm)	2.29	2.08	2.50
Bleeding on probing (%)	13.4	8.0	23.3

**Table 3 ijerph-19-09937-t003:** Comparisons of periodontal epithelial surface area (PESA) or periodontal inflamed surface area (PISA) of the Japanese version and the original version using clinical data (*n* = 210).

	Japanese Version			Original Version ^1^			*p* ^4^
	Median	25th Percentile	75th Percentile	Median	25th Percentile	75th Percentile	
PESA ^2^	1232.3	983.8	1406.7	1049.0	879.1	1203.6	<0.001
PISA ^3^	172.4	84.6	306.3	151.6	75.2	264.9	<0.001

^1^ calculated using the previous study [1]. ^2^ periodontal epithelial surface area. ^3^ periodontal inflamed surface area. ^4^ Wilcoxon signed-rank test.

## Data Availability

The data presented in this study are available on request from the corresponding author. The data are not publicly available due to ethical restrictions.

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
