# Peer review of "Development of a Japanese Version of the Formula for Calculating Periodontal Inflamed Surface Area: A Simulation Study"

_ijerph, 2022, doi:10.3390/ijerph19169937_

Round 1

Reviewer 1 Report

This study developed a Japanese version of the PISA and make a comparison with the original version with stimulation and clinical data.

However, I have some concerns about this study.  The authors used their previous published clinical data to confirm their comparison. However, these participants have a variety of diseases conditions. As mentioned in the study [Ref 8], “We did not account for the medical conditions of participants in this study, because a wide variety of diseases and medications were involved.” As we know, periodontal inflamed surface area (PISA) was associated with systemic conditions. Do these results affect or have a bias effect on the comparison results between the PISA and PESA? Is the sample size sufficient to control for many confounding variables?

Author Response

Thank you for your comments. As you pointed out, the clinical data used in the multiple regression analysis did not include possible confounders. Since the original purpose of the present study was to compare the original and Japanese versions of the PISA, if this shortcoming occurs to the same degree in both models, its impact can be seen as being of equal magnitude and may have little effect on the comparison. This limitation was added in the discussion section (lines 274-280).

We thought that the sample size of the model was sufficient because we planned to use three explanatory variables: age, sex, and PISA. The previous study evaluating the PISA and BMI by multiple regression models using six explanatory variables was based on 71 subjects (Takeda, K.; Mizutani, K.; Minami, I.; Kido, D.; Mikami, R.; Konuma, K.; Saito, N.; Kominato, H.; Takemura, S.; Nakagawa, K.; et al. Association of periodontal pocket area with type 2 diabetes and obesity: a cross-sectional study. BMJ Open Diabetes Res Care 2021, 9). We added this information in the methods section (lines 119-121).

Reviewer 2 Report

Congratulations for the article and the way the methodology was applied

just a correction on line 77

"of a1 through a6 were the coefficients"

Author Response

Thank you for your comment. We corrected “6” to a subscript (line 87).

Reviewer 3 Report

The authors proposed a simplified method for calculating periodontal inflamed surface area in a Japanese population. Per se, this approach is interesting and can be exported to other populations. Nevertheless, there are some issues that need to be addressed before considering for publication.

1. It has to be properly acknowledged how this method represents a rough approximation of the original version proposed by Nesse et al. and mathematical differences between the two methods should be highlighted to improve the reader's understanding (linear function vs polynomial function). 

2. In M&M, it is not clear how the extracted teeth should be representative of the Japanese population. It is preferable to have this information in M&M section: 'the teeth were selected randomly from more than 20,000 extracted teeth collected from 130 dentists 247 across Japan'.

3. In general, M&M section could be better structured. For example, the information regarding the study type, EC approval, etc. and statistical analysis are preferably reported as separated paragraphs. 

4. From the periodontal status (median PPD 2.29 and BoP 13%), it seems that the present population was somehow periodontally healthy. It would have been interesting to perform the same calculation on subjects with more severe periodontitis, since this is the population in which PISA it is more useful for research and clinical purposes. 

5. The present reviewer is not particularly enthusiastic about the statement: 'BMI was relatively more highly correlated with the PISA values from the 214 Japanese version than those from the original version after adjusting for age and sex, suggesting that the PISA values from the Japanese version reflect the amount of periodontal inflammation more precisely than those from the original version'. From the study design realised, this sounds like an overstatement.

Author Response

  1. It has to be properly acknowledged how this method represents a rough approximation of the original version proposed by Nesse et al. and mathematical differences between the two methods should be highlighted to improve the reader's understanding (linear function vs polynomial function).

Response: Thank you for your comment. We added a sentence to make it easier for the readers to understand (lines 216-218).

  1. In M&M, it is not clear how the extracted teeth should be representative of the Japanese population. It is preferable to have this information in M&M section: 'the teeth were selected randomly from more than 20,000 extracted teeth collected from 130 dentists across Japan'.

Response: We added the information in the M&M section, as suggested (lines 72-74).

  1. In general, M&M section could be better structured. For example, the information regarding the study type, EC approval, etc. and statistical analysis are preferably reported as separated paragraphs.

Response: As you suggested, we first explained that this study consisted of three phases (lines 62-68). We also restructured it so that EC approval and statistical analysis are reported in separate paragraphs (lines 106, 113, 124).

  1. From the periodontal status (median PPD 2.29 and BoP 13%), it seems that the present population was somehow periodontally healthy. It would have been interesting to perform the same calculation on subjects with more severe periodontitis, since this is the population in which PISA it is more useful for research and clinical purposes.

Response: We agree with the reviewer. We added this point to the discussion section as a limitation and as needing further study (lines 280-284).

  1. The present reviewer is not particularly enthusiastic about the statement: 'BMI was relatively more highly correlated with the PISA values from the Japanese version than those from the original version after adjusting for age and sex, suggesting that the PISA values from the Japanese version reflect the amount of periodontal inflammation more precisely than those from the original version'. From the study design realised, this sounds like an overstatement.

Response: We revised the relevant sentences to read that the correlations with BMI were similar for both versions of the PISA (lines 22, 233-236, 293-295).

Reviewer 4 Report

This reviewer agrees that PISA is an important diagnostic parameter in determining the severity of periodontitis, and PISA can also predict the extent of various related diseases. Certainly, the current PISA may not be clearly applicable to the Japanese or Asian population. To this end, the authors have developed a Japanese version of the PISA formula. In the future, it will be useful to create PISA formulas for different ethnic groups.

The authors recruited enough numbers of patients (220) and developed a formula for PISA in Japanese.

The manuscript is well written, and all experiments also were well designed and executed.

All the references are appropriately selected.

Author Response

Thank you for your comments.